# MNI: An enhanced multi-task neighborhood interaction model for recommendation on knowledge graph

Xintao Ma[1,2], Liyan Dong[1,2], Yuequn Wang[1,2], Yongli Li[3], Hao Zhang[1,2]*

**1** College of Computer Science and Technology, Jilin University, Changchun, China, **2** Key Laboratory of Symbolic Computation and Knowledge Engineering of Ministry of Education, Jilin University, Changchun, China, **3** School of Information Science and Technology, Northeast Normal University, Changchun, China

* zhangh@jlu.edu.cn

**Data Availability Statement:** The data underlying the results presented in the study are available from Movielens, Book-Crossing, and Last-FM. Movielens:https://grouplens.org/datasets/movielens/. The data is third party and we confirm

## Abstract

To alleviate the data sparsity and cold start problems for collaborative filtering in recommendation systems, side information is usually leveraged by researchers to improve the recommendation performance. The utility of knowledge graph regards the side information as part of the graph structure and gives an explanation for recommendation results. In this paper, we propose an enhanced multi-task neighborhood interaction (MNI) model for recommendation on knowledge graphs. MNI explores not only the user-item interaction but also the neighbor-neighbor interactions, capturing a more sophisticated local structure. Besides, the entities and relations are also semantically embedded. And with the cross&compress unit, items in the recommendation system and entities in the knowledge graph can share latent features, and thus high-order interactions can be investigated. Through extensive experiments on real-world datasets, we demonstrate that MNI outperforms some of the state-of-the-art baselines both for CTR prediction and top-N recommendation.

## 1 Introduction

Nowadays, we are in the era of data explosion as the internet develops rapidly, which raises a lot of obstacles for users to find their interested information. Recommendation systems aim to tackle this problem by assisting users in exploring among massive data and excavating appealing information tailored for each user. Traditional recommendation systems suffer from the problem of data sparsity and cold-start. One approach is collaborative filtering (CF) [1], which cross-compares the users' historical interest and then recommends the items according to their common preferences. However, the recommendation precision relies highly on the sparsity of user-item interactions. Therefore, researchers tend to integrate side information that includes user portraits [2], item attributes [3], text reviews [4] during recommendation.

One popular strategy is to combine the knowledge graph with the recommendation system. A Knowledge graph is a heterogeneous graph, wherein entities and relations are represented by different types of nodes and edges in the graph. Besides, side information can be mapped into a knowledge graph with entities and semantical relations [5, 6]. As shown in Fig 1, the

that others would be able to access these data in the same manner as the authors. We confirm that the authors did not have any special access privileges that others would not have.

**Funding:** The author(s) received no specific funding for this work.

**Competing interests:** The authors have declared that no competing interests exist.

movie "Avatar" and "Blood Diamond" are recommended to the user "Bob". The users, movies, actors, directors, and other attributes are mapped into entities in the knowledge graph. One important contribution of the knowledge graph is that it facilities the recommendation with explainability. For instance, we recommend "Catch me if you can" to Bob because Bob likes "Inception" which shares the same actor "Leonardo DiCaprio". In many tasks, the knowledge graph can serve not only the structure information but also the rich semantics of user-item interactions so that the behavior of users can be better captured. Those tasks such as link prediction [7], natural language processing (NLP) [8], network similarity [9], etc., are fulfilled by knowledge graph embedding (KGE), which projects the entities and relations into a low dimensional space while preserving the important information such as structural and semantical information.

A typical knowledge graph embedding method is translational distance models such as TransE [10] and its extensions [11–13], which by a translation vector they embed the entities and relations into a vector space and evaluate the distance between the vectors. Though those methods are trivial to implement, but some of them lack the semantical understanding of entities and relations. Besides, some researchers are inspired by the development of deep learning algorithms such as graph neural networks. The deep-learning algorithms are widely used in many research fields, such as social networks [14], bioinformatics [15, 16], web advertising [8], etc. For example, based on the idea of graph convolution networks (GCNs) [17, 18], Wang et al. [19] proposed KGCN, which updates the entity representation by its neighbors, then propagates the information to another part of the knowledge graph. Also, inspired by graph attention network [20], KGAT [21] was proposed to model a high-order relation with attention mechanism in knowledge graphs. These methods take into consideration graph structural information as to improve the recommendation precision. Furthermore, Qu et al. [22] proposed knowledge-enhanced neighbor interaction (KNI), which fixes the problem of early summarization problem in existing graph-based models. The early summarization problem is caused by the fact that the models usually compress the neighbor information into the nodes before prediction, making the local structure of those neighbors implicit in utilizing. KNI compresses the user-item interactions and high-order neighbors to increase the graph local connectivity. However, KNI flaws in treating all the relations as one semantic "link exists or not", neglecting the semantical and fruitful facts of relations.

Another trend of knowledge graph embedding is incorporating multi-task learning. Wang et al. proposed multi-task feature learning for knowledge graph and recommendation (MKR) [23], which in one way separates the learning tasks of recommendation system (RS) and the knowledge graph embedding, and in the other way inter-relates the two tasks by the relationship between the items in RS and the entities in KG. This method completes the user-item interaction with entities in the knowledge graph, which improves the recommendation precision. However, it neglects some useful information by the scoring matrix.

To address the limitations of the existing methods mentioned above, we propose an enhanced **m**ulti-task **n**eighborhood **i**nteraction (MNI) model for recommendation on knowledge graph, which utilizes the multi-task learning and also extends the user-item interaction to their neighbors. To be specific, we adopt the idea of KNI to compress the knowledge graph into a local structure where both users' and items' neighbors are collected for prediction tasks. This process avoids the early summarization problem. Then in order to distinguish the different semantic information of relations, we integrate them into item entities and interact with the items in the local structure we build.

In summary, our main contributions of this paper are listed as follows:

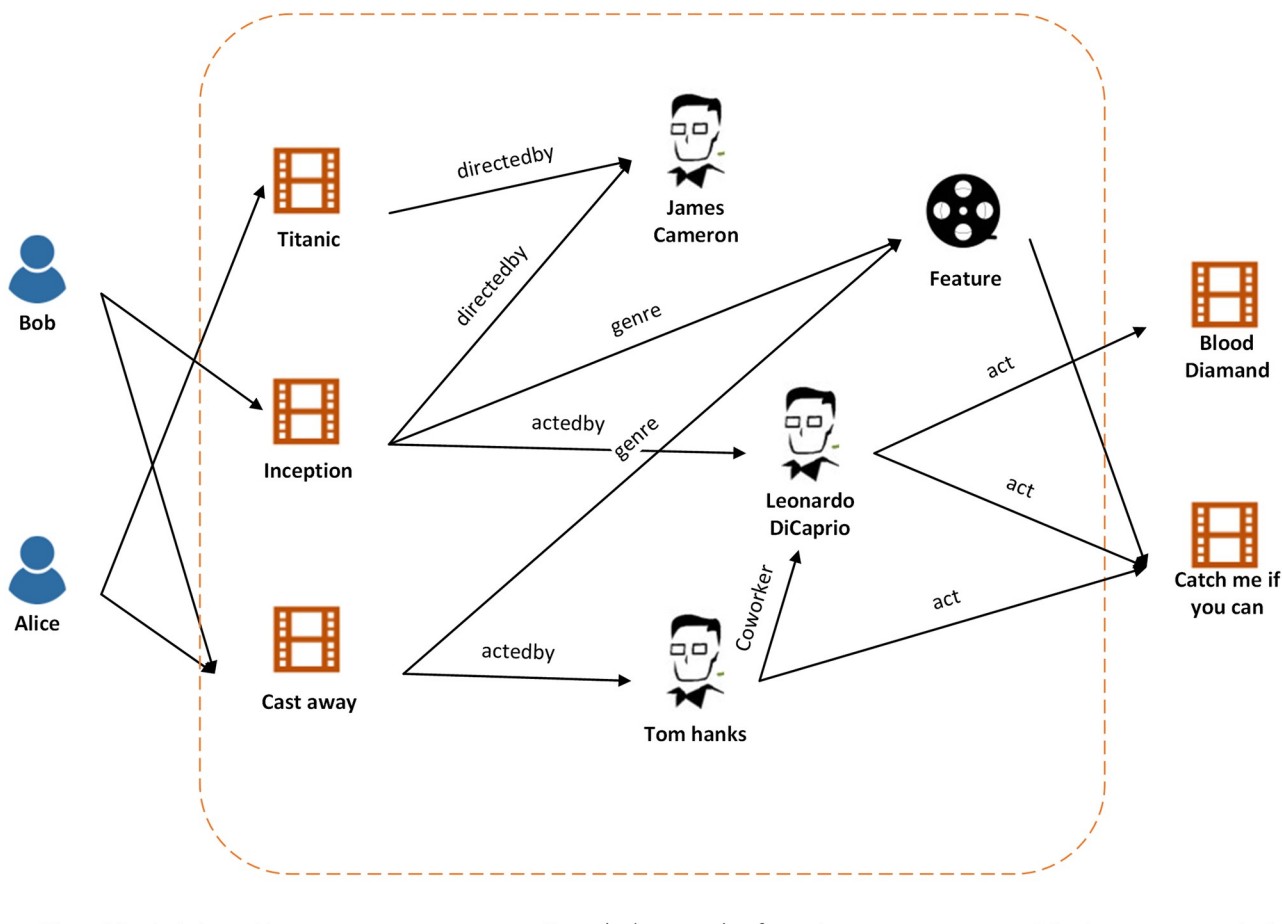

**Fig 1. An illustration of knowledge graph enhanced recommendation system.** The relations between items are annotated with different semantic meanings and the knowledge graph provides detailed facts about movies and their attributes.

- We combine the multi-task learning with enhanced user and item neighbors for preference propagation, which entitles the relations with different semantics and also enhances the recommendation system with neighbor interactions.

- We propose MNI, a framework that reconstructs the knowledge graph to a local neighbor interaction structure, and utilizes the connection between recommendation system and knowledge graph with mutual enhancement to improve the recommendation precision.

- We evaluate our framework with extensive experiments on real-world KG datasets. The results show that our framework achieves better performance compared with other state-of-the-art algorithms.

The rest of this paper is organized as follows: we first introduced some related knowledge graph embedding methods in Section 2. Then we illustrate some notations and explain our framework in Section 3. Next, we demonstrate the experiments with the evaluation metrics and analyze the results in Section 4. Finally, we conclude our work and show some future work in Section 5.

## 2 Related work

Many recommendation techniques have been put forward to solve the data sparsity and cold-start problems in a recommendation system, where side information comes into researchers' sights. One effective way to utilize the side information is factorization machine (FM) [24] and FFM [25]. The idea is to embed those side information into feature vectors, and then integrate them together with the user and item vectors to train together. However, the time of parameter learning and the prediction is linear. Later, DeepFM [26] is proposed that combines the idea of FM and neural networks. The FM module is used to extract the low-order latent features, and the neural network is used to capture the high-order features.

Another effective method is to build the user-item interaction graphs and their side information into nodes and edges, whereby various graph representation learning methods can be applied to learn the low-dimensional embedding to graph vertices. These embeddings preserve the information of graph topology, node similarity and others. Graph convolutional network (GCN) proposed by Kipf et al. [17] adopts transductive learning to learn all the node embedding. Then GraphSAGE [18] is proposed for inductive node embedding. By incorporating node features, GraphSAGE learns the whole graph topology by integrating each node with its neighbors. Then PGE [27] optimized the sampling process by node clustering to assign bias to different neighbors, and then aggregated neighbors information. Besides, graph attention network (GAT) applied the idea of attention mechanism [20], whereby different neighbors are assigned with different weights for information propagation.

In particular, knowledge graphs are widely used to incorporating side information and user-item interactions into graphs. There are three categories regarding KG-based recommendation systems: embedding-based, path-based, and propagation-based methods [27]. Firstly, embedding-based methods directly use the side information to enrich the representation of users or items. Zhang et al. proposed CKE [28] to utilize different types of side information such as textual and visual knowledge into the embedding process. Therefore, CKE combines the structural information and side information of items to improve the recommendation precision. Later CKFG [29] is proposed to constructs a user-item knowledge graph, so that user behavior can also be learned together with the item side information. Besides, Wang et al. proposed SHINE [14], which leveraged the auto-encoder to aggregate the users' social network and their profile with the target item. Wang et.al also proposed MKR [23], which separates the recommendation module and the KGE module. The recommendation module is used to learn the latent representation of users and items, and the KGE module is used to learn the semantic relations and item-related entities. Then those two models are trained through a cross-unit.

Secondly, path-based methods aim to take the connectivity similarity of users or items to enhance the recommendation. Yu et al. proposed Hete-MF [30] and HeteRec [31], which extract meta-paths and the similarity of the paths to learn the representation of users and items. Then Zhao et al. came up with FMG [32] by replacing the meta-path by the meta-graph, which has richer connectivity information. Recently, some studies are carried out to learn the explicit representation of meta-paths. MCRec [33] is designed to learn the interaction context representation from the meta-paths. HetNERec [34] is proposed from another aspect, by constructing the co-occurrence networks to discover the implicit relationship.

The third category is propagation-based methods, which leverage the idea of GCN of information propagation [35]. The first work is Ripplenet by Wang et al. [36], which introduces the concept of ripple set and propagates the user's preference to enrich the user's representation. Contrarily, KGCN [19] is designed by enriching the representation of the candidate item with the embedding of entities and their neighbors in KG. Besides, Wang et al. proposed KGAT [21], aiming to model the high-order relations between users and items using attention-based

information propagation. Recently, KNI is designed by Qu et al. [22], which incorporates the neighbors of items and also the neighbors of users. Thus the refinement of user and item embedding are pulled together. Besides, Wang et al. proposed Ripp-MKR [37], a deep framework that combined the main idea of Ripplenet and MKR. The framework enriches the recommendation system with users' historical clicked-items, and combined with knowledge graph information.

However, our framework differs from the above literature in that: we apply the neighbor interaction of users and items with enhanced multi-task learning, so that the refinement of item representation is the mutual interaction of user-item-neighbor pair and item-entity pair. Unlike Ripp-MKR, our framework focuses on the neighbor-neighbor interactions, which is not limited to user-item interactions. Upon the reconstruction based on the neighborhood interactions of the graph, it can fully discover the implicit relations between items in order to capture the whole structure of the recommendation system. We also apply attention mechanism to focus on the important semantics on edges to improve recommendation precision. The details are explained in Section 3.

## 3 Proposed work

Our framework MNI is shown in Fig 2. It contains three parts: the recommendation module, knowledge graph embedding (KGE) module, and the cross&compress unit. The

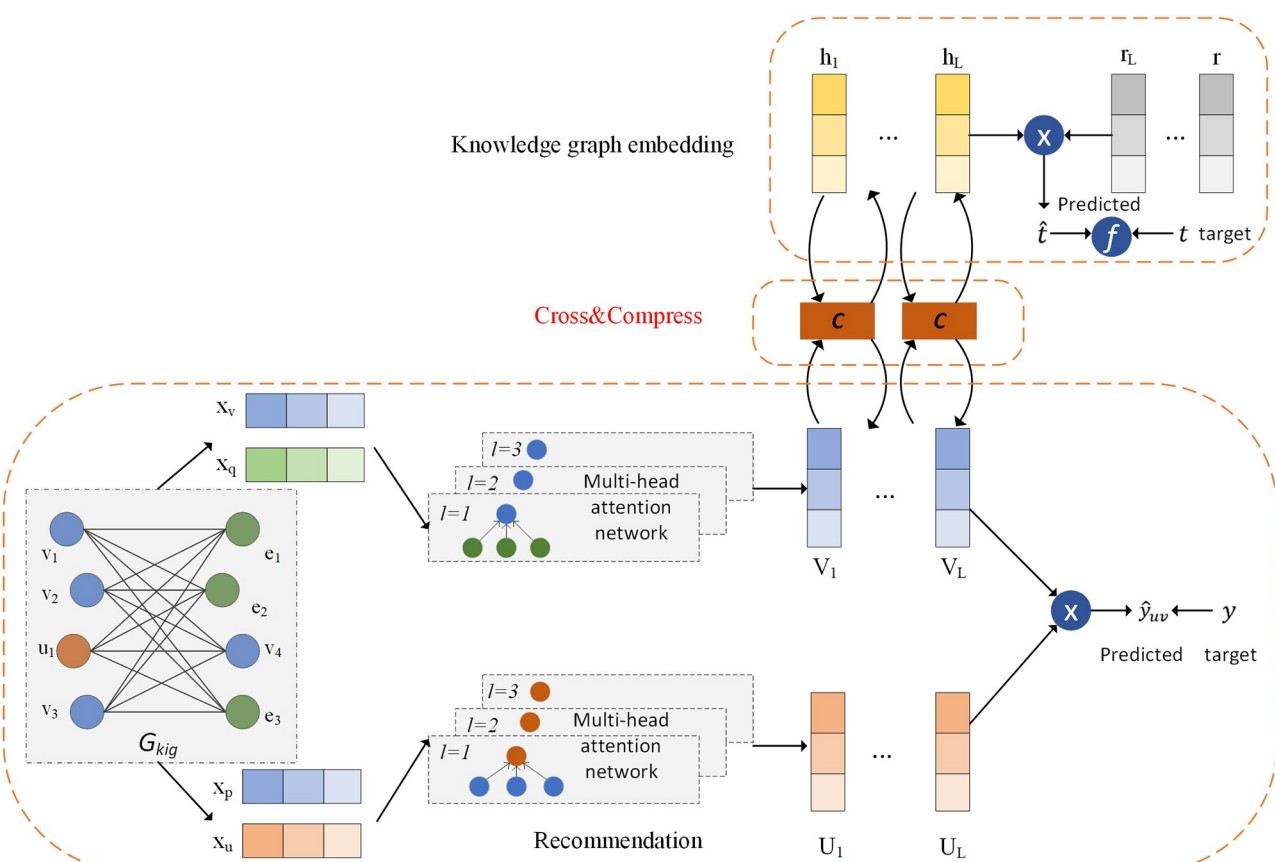

**Fig 2. The framework of MNI.** The bottom part is the recommendation module where the knowledge-enhanced interaction model (KIG) is constructed, and high-order neighborhood integrated by multi-head attention network. The middle part is the cross&compress unit, a bridge for exchanging latent features. The upper part shows the knowledge graph embedding that predicting the target tail entity by head and relation embedding.

**Fig 3. The reconstruction from knowledge graph to knowledge-enhanced interaction model.** The orange circles are users, blue circles are items, and green circles are item attributes. Dash circles denote the user and its neighbors. The process from (b) to (c) demonstrate the high-order neighborhood information is propagated from 2-hop neighbors to 1-hop neighbors. (d) denote the final model, where the edges represent interaction among interactions.

recommendation module is inspired by the work KNI [22]. The first step is to reconstruct the graph into neighbor-interaction graph, and the process is shown in Fig 3. With the reconstructed graph $G_{kig}$, yielding the initial vectors $x_{u,v}$ and their neighbors $x_{p,q}$. We adopt GAT (multi-head attention) to learn the high-order information in the reconstructed graph as shown in the bottom part of Fig 2. Here, the blue blocks represent items, the orange blocks represent users, and the green ones represent entities. Thus, the embedding of users $U_L$ and items $V_L$ can be obtained. The knowledge graph embedding module shown on the upper part of Fig 2, uses a multi-layer to extract the features of head and relation from the triples, which can preserve the semantic of relations and also the graph structure information. The head entity vectors are represented as yellow blocks and the relation vectors are shown as grey blocks. Then with the cross&compress unit [23], a bridge is built between the recommendation module and the KGE module. The bridge exchanges and compensates for the latent information produced by the two modules through items $V_L$ and head entities $r_L$. Afterwards, the predicted user-item interaction values $\hat{y}_{uv}$ and the predicted tail entities $\hat{t}$ are calculated and converged by the loss function. Therefore, our framework aggregates the high-order neighborhood information in the recommendation system and also the semantic relation information in the knowledge graph.

In this section, we will first show some necessary notations and formulations in subsection 3.1. Then we explain our framework module by module, namely the cross&compress unit in subsection 3.2, the recommendation module in subsection 3.3, and the KGE module in subsection 3.4. Finally, we will discuss the learning algorithm in subsection 3.5.

### 3.1 Formulation

The knowledge graph recommendation system is normally formulated as follows. A recommendation system $G_{rs}$ contains the user set $U$ and the item set $V$, represented as $U = \{u_1, u_2, \ldots, u_m\}$ and $V = \{v_1, v_2, \ldots, v_n\}$, where $m$, $n$ are the number of users and items. Thus the user-item interaction can be represented as $Y = \{y_{uv} | u \in U, v \in V\}$. $y_{uv}$ equals to 1 when an interaction between the user $u$ and the item $v$ is observed, such as behaviors like purchasing or rating; otherwise, $y_{uv} = 0$. Besides, a knowledge graph $G_{kg}$ is represented as massive entity-relation-entity triples $(h, r, t)$, which correspond to head entity $h \in E$, relation $r \in R$, and tail entity $t \in E$, where $E$ represents the entity sets and $R$ is the relation set. For instance, (*Inception*, *film.actor*, *Leonardo Dicaprio*) is a triple that states the fact Leonardo is an actor of the film Inception. An entity can form multiple triples and associate with other entities in the KG. The entity Inception can also link to the entity Christopher Nolan with relation film.director.

Thus we can formulate our recommendation task as follows:

- Input: recommendation system $G_{rs}$ and knowledge graph $G_{kg}$

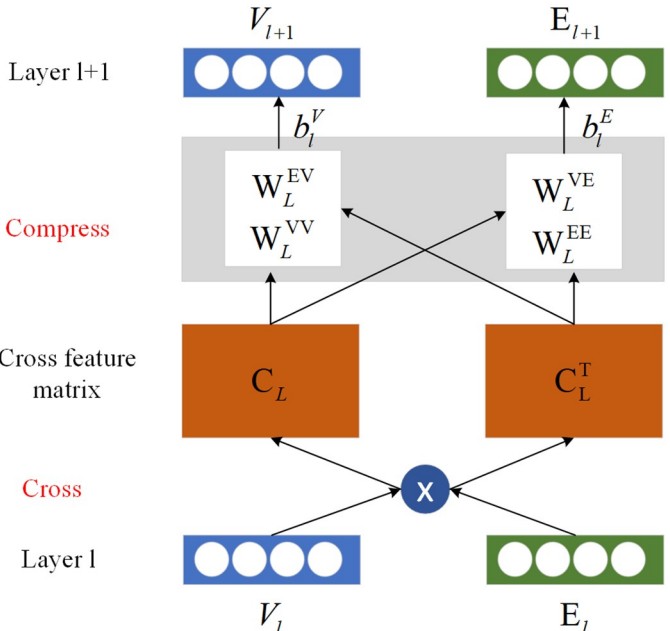

**Fig 4. The illustration of cross&compress unit.** The item and entity vectors are operated by cross function to generate the cross feature matrix.

- Output: a prediction function that predicts the probability $\hat{y}_{uv} = F(u, v)$ that the user $u$ interacts with the item $v$.

## 3.2 The cross&compress unit

The cross&compress unit is mainly designed to build a bridge between the recommendation module and the KGE module [22], through which the two models can share the latent features on each side. The communication is performed between the item and its related entities as shown in Fig 4.

We first introduce the cross feature matrix between the latent feature of item $v_l \in R^d$ and the latent feature of entity $e_l \in R^d$ from layer $l$, which describe the cross function.

$$C_l = v_l e_l^T = \begin{bmatrix} v_l^1 e_l^1 & \cdots & v_l^1 e_l^d \\ \vdots & & \vdots \\ v_l^d e_l^1 & \cdots & v_l^d e_l^d \end{bmatrix} \tag{1}$$

where $C_l \in R^{d \times d}$ is the cross feature matrix of layer $l$, and $d$ is the number of hidden layer. Additionally, the compress function is also performed by projecting the cross feature matrix into the next layer latent representation space:

$$v_{l+1} = C_l w_l^{VV} + C_l^T w_l^{EV} + b_l^V \tag{2}$$

$$e_{l+1} = C_l w_l^{EV} + C_l^T w_l^{EE} + b_l^E. \tag{3}$$

where $W_l^i \in R^d$ and $b_l^i \in R^d$ are the learning parameters. We say that it is a compress function

because this operation projects the matrix from $R^{d\times d}$ space back to $R^d$. To be used later, the cross&compress unit is denoted as the formula:

$$[v_{l+1}, e_{l+1}] = C(v_l, e_l). \tag{4}$$

We use the suffix [v] or [e] to infer to the output separately. Thus the latent features of the item side module and its related entity side module can be exchanged and mutually strengthened. One thing to notice is that the cross&compress unit exists only in the low layer of the framework, because the feature transfer-ability decreases as the layer getting higher [23, 34].

## 3.3 The recommendation module

Most works aggregate the neighbor information before learning the interactions, which compresse the graph structure into two nodes and an edge for prediction. However, this prevents the methods exploring the local structure, which is called early summarization. In [18], the knowledge-enhanced neighborhood interaction (KNI) model is introduced to tackle this problem, where user-item interactions are further explored as neighbor-neighbor interaction and then integrated with high-order neighborhood information.

In this module, we first transform the graph $G = G_{rs} \cup G_{kg}$ to the knowledge-enhanced interaction graph $G_{kig}$.

**Definition 1.**   Given the graph $G = G_{rs} \cup G_{kg}$, where $G_{rs}$ contains the user set $U$ and item set $V$, $G_{kg}$ contains massive triples $(h, r, t)$. Let $N_u$ be the neighborhood of the user, and $N_v$ be the neighborhood of the item. Thus we build $G_{kig} = \{(i, j) | i \in U \cup N_u, j \in V \cup N_v\}$.

We argue that in the newly constructed graph $G_{kig}$, interactions exist between user and item $(u, v)$, user and item neighbor $(u, q)$, user neighbor and item $(p, v)$, user neighbor and item neighbor $(p, q)$, where $p \in N_u$ and $q \in N_v$. Those interactions are then all taken into consideration for prediction. Fig 3 shows the reconstruction of the knowledge-enhanced interaction graph. We can see in Fig 3d, the user neighborhoods and item neighborhoods are collected to compute the neighborhood interactions. We use a bi-attention network to use the information of neighborhood interactions:

$$a_{p,q} = softmax\left( w^T concat\left( x_u, x_p, x_v, x_q \right) + b \right). \tag{5}$$

$$\hat{y}_{uv} = \sum_{p \in N_u} \sum_{q \in N_v} a_{p,q} <x_p, x_q>. \tag{6}$$

where $x_u$ is the embedding of the user $u$, $x_p$ is the user neighbor node $p$, $x_v$ is the item $v$, $x_q$ is the item neighbor node $q$, $<>$ is the inner product. We can see that Eq 1 is different from the attention function in [8, 28, 34], Eq 5 takes the user neighbor node and item neighbor node into account, and then in Eq 6, different interaction assigns different weight according to the attention parameter $a_{i,j}$. This means that the early summation problem is solved by reconstructing the graph into $G_{kig}$, and all types of neighbor interaction are calculated and weighted for prediction.

The next step is integrating high-order neighbor information into the process of prediction. Studies show that high-order neighbor information can reveal latent taste of the user [3, 20, 38], which improves the precision of recommendation. Thus we use graph attention network (GAT) [20] to explore the high-order neighbor information and integrate it into the reconstructed graph. GAT is similar to graph convolution networks except that multi-head self-attention networks are calculated. It allocates different weights to different neighbors, so that the influence of the important neighbors is emphasize and absorbed by the whole graph. Here

we use 2-layer GAT:

$$x_p^1 = \sigma\left(\sum\nolimits_{j \in N_p} a_{p,j}^1 w^1 x_j + b^1\right). \tag{7}$$

$$x_u^2 = \sigma(\sum_{i \in N_u} a_{u,i}^1 w^2 x_p^1 + b^2). \tag{8}$$

where $a_{i,j}^l$ is the attention score of node $i$ to node $j$ in the first layer attention network, $x_p^1$ and $x_u^2$ are the outputs of the first and second attention layers, $w$ and $b$ are the learning parameters weights and bias, and we use Relu for the activation function $\sigma()$. $a_{i,j}^l$ is calculated by:

$$a_{i,j}^l = \frac{\exp(LeakyRelu(w_a^{l\,T} concat\left(x_i^{l-1}, x_j^{l-1}\right) + b_a^l))}{\sum_{k \in N_i} \exp(LeakyRelu(w_a^{l\,T} concat\left(x_i^{l-1}, x_k^{l-1}\right) + b_a^l))} \tag{9}$$

where $w_a^l$ and $b_a^l$ are the learning parameters of the attention network.

Thus, for any node $i$, we can calculate the embedding $x_i^l$ using Eqs 7–9. And then we replace any feature vectors in Eq 6 with the calculated $x_i^l \in G_{kig}$, so that the reconstructed graph contains the high-order neighborhood information, shown as Fig 3. The process can be explained from Fig 3b and 3c that the information of 2-hop neighbors are propagated to the 1-hop neighbors.

Besides, in order to cope with large graphs, we adopt sampling to use only a fixed-number neighbors for attention mechanism.

$$\tilde{N}_i = sample(N_i, k). \tag{10}$$

Here we use neighbor sampling (NS) [18], which randomly samples $k$ neighbors of each node, and thus controls the complexity of the neighborhood information. Other methods can also be used for sophistical sampling, such as random walks [39–41].

Therefore, we can get the embedding for users and items $u_l$ and $x_u^l$ and $x_v^l$ integrating the high-order neighbor information and the interactions among different types of neighbors. In order to align the vectors and extract the feature of users and items, we put them into another fully connected neural network $M^L$ with L layers, Then for an item, we use $L$ cross&compress units to combine the feature of the item and the related entities.

$$U_L = M^L\left(x_u^l\right). \tag{11}$$

$$V_L = E_{e \sim S(v)}[C^L(v, e)[M^L(x_v^l)]]. \tag{12}$$

where $M(x) = \sigma(W_x + b)$ is the fully-connected network, and $S(v)$ is the related entities of item $v$.

## 3.4 The KGE module

In this module, the entities and relations are embedded in a way that the graph structure is maintained and the semantic of relations as well [42]. Since in the recommendation module, the high-order neighborhood information is integrated only by the semantic of existing or not, the information of semantic relation is vacant at some level. Thus, we need the KGE module to preserve the semantic relation in the KG. We use a deep semantic matching model for KGE

embedding.

$$h_L = E_{e \sim S(h)}[C^L(v, h)[e]].$$ (13)

where $C^L$ is the cross&compress unit to output the latent feature of head entity $h$ of layer $L$, $S(h)$ is the related item of the entity $h$.

$$r_L = M^L(r)$$ (14)

$$\hat{t} = M^K\left(\begin{bmatrix} h_L \\ r_L \end{bmatrix}\right)$$ (15)

where the raw vectors of $h$ and $r$ can be feature vectors including ID, types, or textual descriptions. And then their latent features are concatenated to predict the tail entity.

Finally, the score of predicting the tail entity is calculated by the score function $f_{KG}$ shown as follows:

$$score(h, r, t) = f_{KG}(t, \hat{t})$$ (16)

where we use $f_{KG} = \sigma(t^T, \hat{t})$.

## 3.5 Learning algorithm

The loss function of our framework contains three parts:

$$L = L_{RS} + L_{KGE} + L_{REG}$$ (17)

The first loss is the recommendation loss is calculated as:

$$L_{RS} = -\sum_{y_{u,v}=1} \log(\hat{y}_{uv}) - \sum_{y_{u,v}=0} \log(1 - \hat{y}_{uv})$$ (18)

And for the KGE loss, the same as the recommendation loss, we want to increase the score for all positive prediction.

$$L_{KGE} = -\lambda_1 \left( \sum_{(h,r,t) \in G_{kg}} score(h, r, t) - \sum_{(h,r,t) \notin G_{kg}} score(h', r, t') \right)$$ (19)

The last loss term is regularization term that prevents overfitting:

$$L_{REG} = \lambda_2 ||W||_2^2$$ (20)

In conclusion, the whole learning algorithm is shown in Algorithm 1, which we use negative sampling for the training, shown as follows:

```
Algorithm 1. The learning algorithm of MNI
Input: Interaction matrix Y, knowledge graph G.
Output: Prediction F(u, v|Θ, Y, G)
1: Initialize the parameters
2: Reconstruct the graph to knowledge-enhanced interaction graph G_kig
3: For i = 1,..., max iter do
4:   For t steps do
5:     Sample minibatch of positive and negative interactions from G_kig
6:     Sample e~S(v) for each item in the minibatch.
7:     Calculate the gradients and update the parameters on the mini-
batch according to Eqs 1-12, 17 and 20
8:   end for
```

```
 9:     Sample minibatch of true and false triples from G_kg
10:      Sample v~S(h) for each entity in the minibatch.
11:      Calculate the gradients and update the paramters on the mini-
batch according to Eqs 13-16, 19 and 20
12:end for
```

The training process starts with the reconstruction of the graph into $G_{kig}$. Then the process is divided into two procedures: from line 4–7 the recommendation module is trained with the cross&compress unit; line 8–11 represents the training of KGE module. The gradient of loss is calculated as multi-task training with respect to model parameters Θ. One thing to notice is that the recommendation module is trained for $t$ times, we will discuss the choice of t later in the experiment session.

## 4 Experiments

In this section, we will show the performance of our framework on three real-world datasets, and compare the results with state-of-the-art baselines, then take parameter sensitivity experiments.

### 4.1 Dataset

We implement three common real-world datasets: Movielen-1M, Book-Crossing, Last.FM, which are widely used in the field of movies, books, and music. The details are shown as follows:

- **Movielens-1M** [43] is a stable benchmark dataset from Grouplens Research. The data was collected over various time periods, which contains the user profile, movie attributes and ratings. The ratings are from 1 to 5. Here we set positive threshold as 4.

- **Book-Crossing** [44] contains ratings from 0 to 10 of books in the Book-Crossing community, which also contains some attributes of the user and the book.

- **Last-FM** [45] includes the musician listening information from over 2000 users from Last-fm online music system, and also some demographics information of users, tags of tracks.

We use Microsoft Satori [46] to construct the knowledge graph for each dataset. For the three datasets Movielens-1M, Book-Crossing, and Last-FM, we first sample a subset of triples from the knowledge graph, where the relation names contain "movie" or "book" distinguish and also the confidence level is more than 0.9. Then with the subset, we gather all the IDs by matching the names with the tails of the triples, such as (*head, film.name, tail*). Then we match the IDs with the head and tail of all the KG triples, and extend the set of entities to 4 hops iteratively. One thing to notice is that the items without any matching entity are excluded in our experiments. The details of the datasets are listed in Table 1.

**Table 1. The details of the three real-world datasets.**

| Datasets | Movielens-1M | Book-Crossing | Last-FM |
|---|---|---|---|
| Users | 6,036 | 17,860 | 1,872 |
| Items | 2,347 | 14,910 | 3,846 |
| Interactions | 753,772 | 139,746 | 42,346 |
| Triplets | 20,195 | 19,793 | 15,518 |

## 4.2 Baselines

We compare our results with some state-of-the-art algorithms, which are categorized in factorization machines, embedding-based methods, path-based methods, and propagation-based methods of knowledge graph embedding. They are listed as follows:

**LibFM** [47]: It is a widely used factorization machine model. We use TransR [11] to pretrain the graph and obtain the entity embedding as input to LibFM, where the dimension of TransR is 32. For the user and item input we use raw feature vectors. Here the dimension is {1,1,8} and the training epoch is 50.

**Wide&Deep** [48]: It is another feature-based FM model, which uses deep learning models and the shallow models. The input here is the same as LibFM as described above. The dimension of user, item, and entity is 64. And a two-layer channel is processed with the dimension of 100 and 50.

**CKE** [28]: It is a representative of embedding-based method for KG recommendation, which embeds the semantic side information by TransR such as texual, image, and graphs information. The dimension of entity is 32, and the user and item for the three datasets are 64, 128, 32.

**DKN** [8]: It is another representative embedding-based method for KG recommendation. DKN utilizes multiple channels to embed the entity as well as word, and then combines them in a CNN for prediction. Here, we use movie and book names as textual input for DKN. The dimension for word and entity embedding is 64, and the number of filters is 128 for window size 1, 2, 3.

**PER** [49]: It is from the category path-based method by treating KG as heterogeneous information networks and extracting the meta-path. We manually set the meta path pattern as [19].

**KNI** [22]: This is a typical propagation-based method by regarding the interaction happen between the neighbors of user and the neighbors of items. Here the hop number is 4 and hidden dimension is 128.

**MKR** [23]: It is the main base algorithm for our framework which mutual trains the user-item pair and entity-relation pair. We set the high-level layer as $K = 1$. The dimension for each data is 8, 8, 4, and the step is 3, 2, 2.

## 4.3 Experiment setup

We evaluate those models on 2 tasks, click-through rate (CTR) prediction and top-N recommendation. For CTR, we use the evaluation metrics Area Under Curve (AUC) and Accuracy (ACC). And for the recommendation task, we use the metrics of Precision@K and Recall@K, which select K items with the highest predicted probability for each user.

Besides, we use the hyper-parameter k = 1 as high-level layers, $\lambda_2 = 10^{-6}$ as the regularization term parameter. For three datasets, we set $t$ as 3, 2, 2. Also, we use hop number of 4 to build the high-order neighborhood. Also, we use 80% of each data for training and the remaining for testing. Within the 80% training set, we randomly choose 20% data for validation.

## 4.4 Results and discussions

In this subsection, we show the comparison results with other baselines, and then discuss over some parameter sensitivity experiments.

**Table 2. Comparison results of AUC, ACC.** The result of LibFM, Wide&Deep, CKE, PER, DKN is taken from [23].

| Model | Movielens-1M | | Book-Crossing | | Last-FM | |
|---|---|---|---|---|---|---|
| | AUC | ACC | AUC | ACC | AUC | ACC |
| LibFM | 0.892 | 0.812 | 0.685 | 0.640 | 0.777 | 0.709 |
| Wide&Deep | 0.898 | 0.820 | 0.712 | 0.624 | 0.756 | 0.688 |
| CKE | 0.801 | 0.742 | 0.671 | 0.633 | 0.744 | 0.673 |
| DKN | 0.655 | 0.589 | 0.622 | 0.598 | 0.602 | 0.581 |
| PER | 0.710 | 0.664 | 0.623 | 0.588 | 0.633 | 0.596 |
| MKR | 0.917 | 0.843 | 0.734 | 0.704 | 0.797 | 0.752 |
| KNI | **0.944** | **0.872** | **0.772** | **0.706** | **0.823** | **0.774** |
| **MNI** | 0.951 | 0.879 | 0.781 | 0.707 | 0.825 | 0.777 |

**Overall comparison.** The comparison results with the baselines regarding CTR prediction is shown in Table 2, and the top-K recommendation results compared with MKR and KNI are shown in Figs 5–7. We can conclude the following observations:

- Our framework performs the best compared with other baselines. Specifically, compared with the two algorithms MKR and KNI that our framework mainly based on, MNI increases the AUC and ACC on all three real-world datasets. On Book-Crossing, our framework increases the AUC by 1.17% compared to KNI and 10.9% compared to MKR. The reason is that our framework takes the local neighborhood structures into account, especially the neighbor interaction integrates all types of neighbors. Besides, by semantically embedding the entities in the knowledge graph and exchanging latent features with the related items full of neighbor information, the embedding is improved by mutual perfection. As for top-K recommendation, our framework also outperforms MKR and KNI, which is consistent with CTR prediction.

- PER's performance is not as ideal as others, because it is path-based method for KG recommendation, which relies highly on the property of meta-paths and requires much human expertise. PER requires much more effort on manually designing the meta-path.

- KNI performs best among all other baselines, which highlights the importance of local neighbor structure [22]. However, KNI treats the interactions among the users, items, and

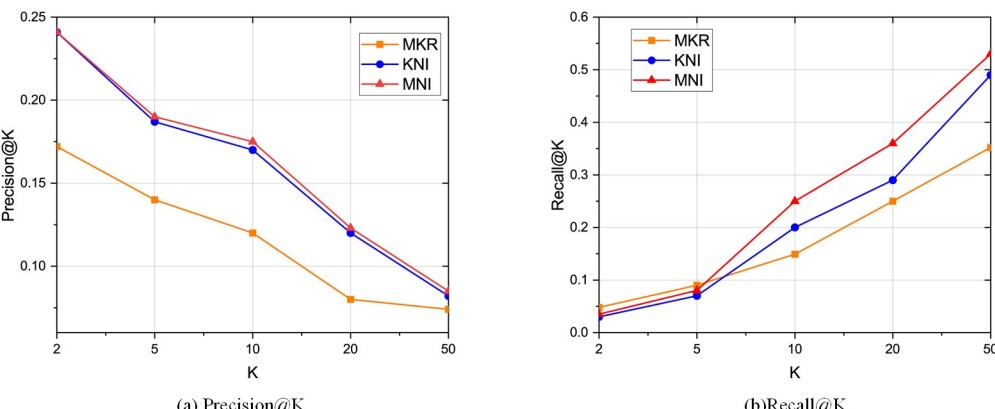

(a) Precision@K        (b)Recall@K

**Fig 5. Top K recommendation result for Movielens-1M regarding precision and recall.**

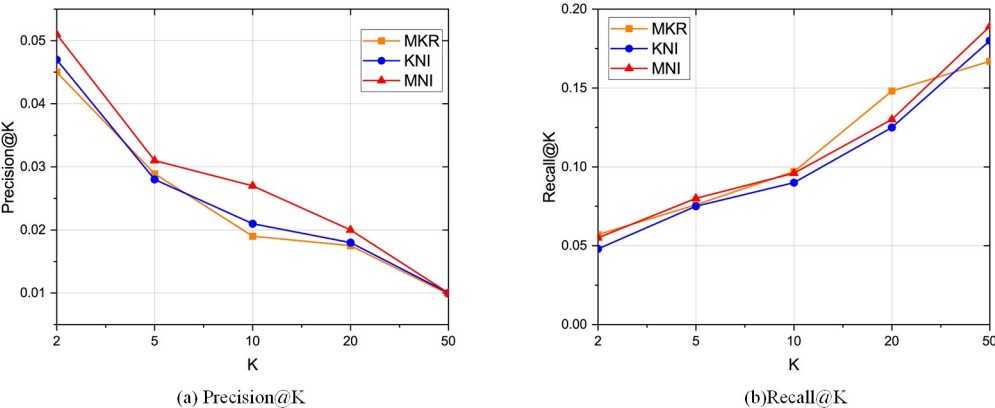

**Fig 6. Top K recommendation result for Book-Crossing regarding precision and recall.**

neighbors as the same, which lacks the semantic meaning of KG relations that also plays an important part in user preference.

- Movielens-1M has the best experiment results because the data is denser. On the other side, Book-Crossing has the worst data sparsity. However, we improve the recommendation results on all datasets, meaning that our framework can deal with data sparsity.

- Propagation-based methods perform better than embedding-based methods and path-based methods. The reason is the propagation methods combine the advantages of those two other types of methods. The result normally contains high-order information and is also explainable. However, some of them are computationally costly as the graph grows large.

- Besides, LibFM and Wide&Deep generally perform better than embedding-based methods, which shows the effectiveness of FM-based methods, especially dealing with sparse data. One reason may be that some of the FM-based models exploit high-order layers, which may explore the high-order information. On the other side, embedding-based methods focus on the explicit interaction between users and items.

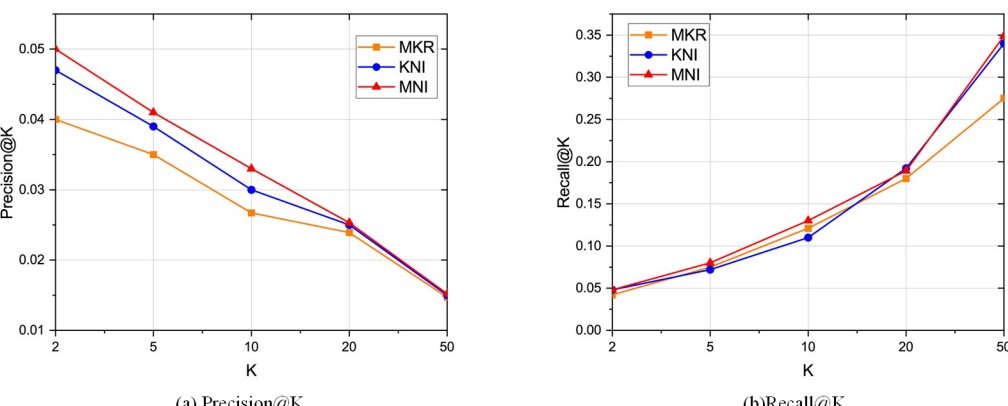

**Fig 7. Top K recommendation result for Last-FM regarding precision and recall.**

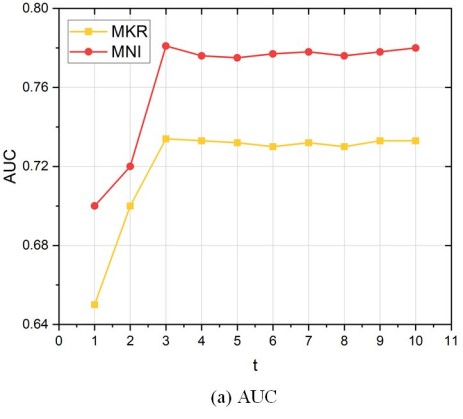 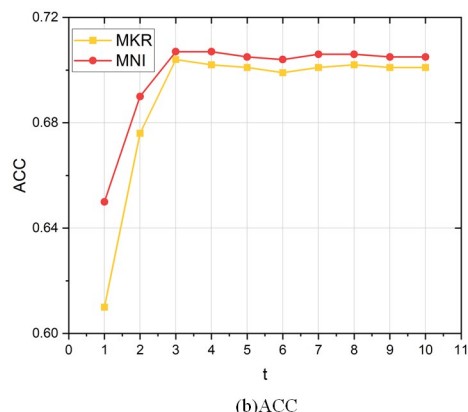

(a) AUC                                                                 (b)ACC

**Fig 8. Training frequency on book-crossing.**

**Table 3. Experimental results on AUC of the effect of hop number.**

| Hop number | Movielens-1M | Book-Crossing | Last-FM |
|---|---|---|---|
| 1-hop | 0.925 | 0.739 | 0.780 |
| 2-hop | 0.937 | 0.756 | 0.791 |
| 3-hop | 0.951 | 0.779 | 0.816 |
| 4-hop | 0.949 | 0.781 | 0.825 |

- For the top-K recommendation results, recall@K increases as K becomes large. One thing to notice is that even when k is very small, our framework still much better than the others, which indicates that our framework can capture the user preference.

**Parameter variation.**   We evaluate our framework on the parameter sensitivity by varying the training step $t$ from 1 to 10. The result is shown in Fig 8. We observe that our framework achieves the best when $t = 3$. The reason is that a high training frequency of the KGE module falsely directs the objective function. On the other hand, a small training frequency cannot combine the transferred knowledge from both sides of the graphs.

Then we tune the hop number of neighbors included in the recommendation module from 1 to 4. The result is shown in Table 3. We can see that the best AUC results on Movielens-1m exist using 4 hop neighbors, and for datasets Book-Crossing and Last-FM, the best results occur at hop = 3. This may because that too much high-order neighbor information introduces noise during training, especially when low-order already includes those neighbors that are also covered by high-order propagation, which is a consistent result with other studies [21, 35].

## 5 Conclusion

In this paper, we propose MNI, which is a multi-task learning approach that incorporates the neighbor-neighbor interaction and the semantical embedding. In the first learning task, we reconstruct the knowledge graph to a knowledge-enhanced neighborhood interaction model, which contains user and item, user and item-neighbor, user-neighbor, and item, user-neighbor and item-neighbor interactions before prediction so that early summarization problem is taken care of. Besides, high-order neighborhood information is also integrated by graph

attention networks. The other learning task is to embed the entities and relations semantically in the knowledge graph. Meanwhile, the item embedding and the head entity embedding exchange latent features through a cross&compress unit, so that knowledge can mutual flows to each side of the task. We conduct a massive experiments on three real-world datasets. The results show that our framework improves the recommendation quality and efficiency.

For future works, we plan to investigate other types of sampling in the recommendation module, such as sampling a subgraph [50]. Random walks can be adopted to generate a walking sequence with most frequent visited nodes, which can emphasize the important neighbors in the recommendation module. Also, another possible improvement is to alter KGE methods to extract more knowledge from the knowledge graph, such as considering implicit relations between items using propagation techniques [21].

## Acknowledgments

We would like to thank Hao Zhang (Jilin University) for the insightful comments on the manuscript and his guidance and patience enlighten us not only on this paper but also our future. We gratefully acknowledge the valuable contribution by Liyan Dong(Jilin University) during preparing the paper.

## Author Contributions

**Conceptualization:** Xintao Ma.

**Data curation:** Xintao Ma, Liyan Dong.

**Formal analysis:** Xintao Ma, Liyan Dong.

**Investigation:** Xintao Ma, Yuequn Wang.

**Methodology:** Xintao Ma.

**Project administration:** Xintao Ma.

**Resources:** Xintao Ma, Liyan Dong, Yuequn Wang, Yongli Li.

**Software:** Xintao Ma, Liyan Dong, Yongli Li.

**Supervision:** Liyan Dong, Hao Zhang.

**Validation:** Xintao Ma, Yongli Li.

**Visualization:** Xintao Ma, Yongli Li.

**Writing – original draft:** Xintao Ma.

**Writing – review & editing:** Xintao Ma, Hao Zhang.

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
