## [Decision Letter · Decision Letter 0]

7 Sep 2021

PONE-D-21-25428MNI: An enhanced multi-task neighborhood interaction model for recommendation on knowledge graphPLOS ONE

Dear Dr. Hao Zhang,

Thank you for submitting your manuscript to PLOS ONE. After careful consideration, we feel that it has merit but does not fully meet PLOS ONE’s publication criteria as it currently stands. Therefore, we invite you to submit a revised version of the manuscript that addresses the points raised during the review process.

We look forward to receiving your revised manuscript.

Kind regards,

Qi Zhao

Academic Editor

PLOS ONE

Journal Requirements:

2. Thank you for your submission to PLOS ONE. Before we can proceed, we kindly ask you to address the following concerns:

Upon internal assessment of your manuscript, we found that it is similar to one of your previous works published in PLOS ONE: "Multitask Feature Learning Approach for Knowledge Graph Enhanced Recommendations with RippleNet".

PLOS policy specifies that if a submitted study is similar to previous work, as appears to be the case in this instance, authors should provide a sound scientific rationale for the submitted work and clearly outline how the new submission differs from the past work. The manuscript should also include, if appropriate, a discussion of the current study in context of previous work. We note that the previous work was not cited in your list of references, nor was it included in the section "Related Works". We would expect it to be included in both of these sections.

We also note that numerous citations in your list of references are conference proceedings. While we do allow for this, we would expect the majority of your statements to be supported by reference to peer-reviewed literature. As such, we suggest that you support your statements with a greater number of peer-reviewed citations.

Furthermore, you have used LastFM, MovieLens and Book-Crossing datasets; however, we ask you to provide the URLs for these sites, and to additionally provide the search terms or other details needed to extract the exact data used in the study. The manuscript should provide sufficient details such that any reader could readily replicate the results of your study. Without details on how you accessed the exact data, this is not possible.

We appreciate your attention to these queries and look forward to your response.

Reviewers' comments:

Reviewer's Responses to Questions

**Comments to the Author**

1. Is the manuscript technically sound, and do the data support the conclusions?

Reviewer #1: Yes

Reviewer #2: Yes

2. Has the statistical analysis been performed appropriately and rigorously? 

Reviewer #1: Yes

Reviewer #2: Yes

3. Have the authors made all data underlying the findings in their manuscript fully available?

Reviewer #1: Yes

Reviewer #2: Yes

4. Is the manuscript presented in an intelligible fashion and written in standard English?

Reviewer #1: Yes

Reviewer #2: Yes

5. Review Comments to the Author

Reviewer #1: This paper describes a recommendation system on knowledge graph, which deploys neighbor-neighbor interactions to explore high-order relations. Then the algorithm applies a cross & compress unit to combine the recommendation system and knowledge graph, whereby those two modules can interact and share latent features to improve the recommendation precision.

Pros:

The algorithm explains well understood and the experiment supports the algorithm assumption and shows improvement compared with other baselines.

Cons:

1.Figure 2 is the whole structure of the framework, however, needs more explanations in the paragraph.

2.There are some mistakes of English grammar in the abstract and the introduction part, needs to correct them. For example, Therefore, researchers tend to integrate side information which includes user… “which” should be changed to “that”

3.The future work should be discussed in detail.

4.The caption of Figure 1 is not self-explained. The caption should state which one is user, item, or attribute.

5.In the experiment, Figure 5-7 the yellow line is not clear enough to recognize. Also, the captions should be in more detail.

Reviewer #2: The algorithm MNI adopts the separation of recommendation system and knowledge graph, and use a bridge unit to share each latent feature. It also applies neighbor-neighbor reconstruction instead of user-item interaction, containing more information about user-user interaction, item-item interaction, and a deep understanding of the high-order relations.

However, some minor correction should be made:

1. Reference format is not consistent. The authors should double check and revise it.

2. The algorithm 1 on page 10 should be explained more in detail. It is fuzzy when just looking at the algorithm.

3. The blocks and colors in the figure 2 is not well explained. Are there any different meaning about different shapes and colors? The author needs to be clearer.

4. About the recommendation module, the author use 2-layer GAT. Could you explain GAT as for the reader is hard to follow.

5. Minor typos, e.g., Missing space between citation and text.

6. PLOS authors have the option to publish the peer review history of their article (what does this mean?). If published, this will include your full peer review and any attached files.

Reviewer #1: **Yes: **Zhen Liu

Reviewer #2: No

---

## [Author Response · Author response to Decision Letter 0]

12 Sep 2021

Dear editor,

I have uploaded the responses in three separate files, response to editors, response to reviewer 1 and 2.

Response to editor:

Q1: Style requirements

Response: We have changed the style and added the line number. The title page and the main body are adapted and hopefully consistent with your requirements. If there is something wrong, please contact us and we appreciate your time.

Q2: Similarity to one of your previous works published in PLOS ONE: "Multitask Feature Learning Approach for Knowledge Graph Enhanced Recommendations with RippleNet":

Response: Sorry for the fuzziness of the two works. Both works are improvements based MKR, however, in two different direction. This new submission differs from the previous one on three aspects:

 We use neighbor-neighbor interactions to explore the recommendation system, which includes user-item interactions, item-item interactions, item-entity interactions, and user-item interactions as illustrated on Page 8 the recommendation module part. This algorithm leads to more accurate recommendation precision. However, in the previous work, it only consider the interactions between users and historical clicked items using Ripplenet. 

 We use a reconstruction step to reformulate the recommendation module to G_kig instead of normal recommendation system in the previous work.

 We leverage attention mechanism to highlight the important edges between neighbors, in order to keep more meaningful semantics.

Thank you for your advices, we also added additional clarification and the comparison between them in the Related work with highlighted texts and also cited the previous work at [37] in the reference in the manuscript. 

Q3: Data access:

Response: Thank you for your suggestion. We added URLs footprints to the data sets of LastFM, Movielens and Book-Crossing on Page 11. Those data can be directed downloaded from the websites and used to replicate our work. But we can also upload our data to the website if requested.

Q4: Peer-reviewed references:

Response: Thank you for your helpful advice to improve the quality of our paper. We add more journal peer-reviewed references and also change some citations’ form. The added reference number are: 9, 13, 14, 15, 16, 34, 37, 41, 50.

Q5: English revision:

Response: We have checked the language, grammar, and spelling by our colleagues, named Tianming Zhao and Jifen Tao. They have made changes to the grammar and the sentence fluency.

Response to reviewer 1:

Point 1: Figure 2 is the whole structure of the framework, however, needs more explanations in the paragraph  

Response 1: Thank you for your advice. We rewrite the first paragraph in section 3 on page 5. We add a detailed procedure for recommendation module and knowledge graph embedding module, pointing out the meaning of symbols and also the color of the blocks. 

Point 2: There are some mistakes of English grammar in the abstract and the introduction part, needs to correct them. For example, Therefore, researchers tend to integrate side information which includes user… “which” should be changed to “that”

Response 2: Thank you for your suggestions. We found it very useful to improve our paper. We changed the English grammar carefully in the abstract and the introduction part.

Point 3: The future work should be discussed in detail.

Response 3: We have adopted your suggestion by extending the future works to random walks and implicit relations. We add some discussion to fulfil the future works and it is clearer to understand.

Point 4: The caption of Figure 1 is not self-explained. The caption should state which one is user, item, or attribute.

Response 4: Thank you for your suggestions. We add some explanation about the Figure 1. 

Point 5: In the experiment, Figure 5-7 the yellow line is not clear enough to recognize. Also, the captions should be in more detail

Response 5: Thank you for your suggestions. We have changed the yellow line to orange to be clearer. Also, we add some modifications in the captions to be detaied.

Response to reviwer 2:

Point 1: Reference format is not consistent. The authors should double check and revise it

Response 1: Thank you for your suggestions. We have modify the reference format so that they are consistent.

Point 2: The algorithm 1 on page 10 should be explained more in detail. It is fuzzy when just looking at the algorithm.

Response 2: We are grateful for your suggestions. We have added some more explanations to Algorithm 1 so that it is clearer. We explain the whole procedure in steps: reconstruction, recommendation module, cross&compress unit, and KGE module.

Point 3: The blocks and colors in the figure 2 is not well explained. Are there any different meaning about different shapes and colors? The author needs to be clearer.

Response 4: Thank you very much for your comments. We have added some explanations about the colors and shapes in the figure 2 in the first paragraph under Section 3. Also, we modify some explanations about Figure 2 to be specific.

Point 4: About the recommendation module, the author use 2-layer GAT. Could you explain GAT as for the reader is hard to follow.

Response 5: We agree with the reviewers comments. We added some explanations about how the GAT works right before the phrase “2-layer GAT”. 

Point 5: Minor typos, e.g., Missing space between citation and text

Response 5: We agree with the reviewers comments.We have double checked through the paper and change the space problem.

---

## [Decision Letter · Decision Letter 1]

27 Sep 2021

MNI: An enhanced multi-task neighborhood interaction model for recommendation on knowledge graph

PONE-D-21-25428R1

Dear Dr. Hao ZHANG,

We’re pleased to inform you that your manuscript has been judged scientifically suitable for publication and will be formally accepted for publication once it meets all outstanding technical requirements.

Kind regards,

Qi Zhao

Academic Editor

PLOS ONE

Additional Editor Comments (optional):

Reviewers' comments:

Reviewer's Responses to Questions

**Comments to the Author**

1. If the authors have adequately addressed your comments raised in a previous round of review and you feel that this manuscript is now acceptable for publication, you may indicate that here to bypass the “Comments to the Author” section, enter your conflict of interest statement in the “Confidential to Editor” section, and submit your "Accept" recommendation.

Reviewer #1: All comments have been addressed

Reviewer #2: (No Response)

2. Is the manuscript technically sound, and do the data support the conclusions?

Reviewer #1: Yes

Reviewer #2: (No Response)

3. Has the statistical analysis been performed appropriately and rigorously? 

Reviewer #1: Yes

Reviewer #2: (No Response)

4. Have the authors made all data underlying the findings in their manuscript fully available?

Reviewer #1: Yes

Reviewer #2: (No Response)

5. Is the manuscript presented in an intelligible fashion and written in standard English?

Reviewer #1: Yes

Reviewer #2: (No Response)

6. Review Comments to the Author

Reviewer #1: The authors have expounded the opinions I put forward in the last round of review, and the data description in the manuscript can also technically support the author's conclusions. It is hoped that the authors will check the English expression before the official publication of the manuscript.

Reviewer #2: (No Response)

7. PLOS authors have the option to publish the peer review history of their article (what does this mean?). If published, this will include your full peer review and any attached files.

Reviewer #1: No

Reviewer #2: No

---

## [Editor Report · Acceptance letter]

19 Oct 2021

PONE-D-21-25428R1 

MNI: An enhanced multi-task neighborhood interaction model for recommendation on knowledge graph 

Dear Dr. Zhang:

I'm pleased to inform you that your manuscript has been deemed suitable for publication in PLOS ONE. Congratulations! Your manuscript is now with our production department. 

Kind regards, 

on behalf of

Dr. Qi Zhao 

Academic Editor

PLOS ONE